# Description of a new low-cost and open-source audiometer and its validation with normal-hearing listeners: The Aupiometer

**Vincent Isnard**📷*, **Véronique Chastres, Guillaume Andéol**📷

Département Neurosciences et Sciences Cognitives, Institut de Recherche Biomédicale des Armées, Brétigny-sur-Orge, France

* vincent.isnard@def.gouv.fr

## Abstract

Hearing loss is a major public health problem. In 2050, it could affect 2.5 billion people. It has therefore become necessary to prevent and diagnose them as early and as widely as possible. However, the costs of clinical equipment dedicated to the functional exploration of hearing remain high and hamper their distribution, while the technologies used are relatively basic. For example, the gold-standard pure-tone audiometry (PTA) essentially consists of emitting pure sounds. In addition, clinical audiometers are generally limited to PTA or few audiological tests, while hearing loss induce multiple functional deficits. Here, we present the Aupiometer, a low-cost audiometer implemented on a modular open-source system based on Raspberry Pi, and which integrates the entire technical framework necessary to carry out audiological measurements. Several hearing tests are already implemented (e.g. PTA, speech audiometry, questionnaires), while the clinical validity of the Aupiometer was verified on a panel of participants (N = 16) for an automated test of standard and extended high-frequency PTA, from 0.125 to 16 kHz, in comparison with a clinical audiometer. For this comparison between the two devices and over this wide frequency range, the difference is evaluated as less than ±10 dB for a 90% confidence interval, of the same order of magnitude as on test-retest differences on a single device. The interest of this device also extends to academic research as it should encourage the prototyping of innovative hearing tests by the community, in order to better understand the diversity of hearing problems in the population.

## Introduction

The number of people with some degree of hearing loss is increasing every year and is expected to reach 2.5 billion people in 2050, of whom 700 million will need hearing rehabilitation. The related costs are estimated at USD$980 billion each year [1]. Exposure to continuous noise represents one of the main causes, due to prolonged exposure to environmental or professional noise (e.g. road traffic, industrial machines [2]) or recreational noise (e.g. bars, concerts [3]). Impulse noises can also generate significant auditory after-effects (e.g. fireworks, sports

**Data Availability Statement:** https://doi.org/10.5281/zenodo.11033174.

**Funding:** The author(s) received no specific funding for this work.

**Competing interests:** The authors have declared that no competing interests exist.

shooting, military service [4]), as can the use of ototoxic medications [5]. The consequences of hearing loss on daily life are particularly harmful, such as desocialization, fatigue, and cognitive decline [6–8]. Their prevention and diagnosis therefore represent a major public health issue.

However, access to hearing assessment may be limited due to a lack of health personnel, for example in rural areas far from medical centers, or in developing countries where access to expensive medical equipment is limited [9]. In particular, Lie et al. [2] note in their meta-analysis that the harmful effects of exposure to noise at work are felt in developing countries, where it is therefore all the more necessary to increase hearing prevention and monitoring. In this context, hearing assessment devices that are more easily accessible to the population are being developed, such as online tests developed by hearing aid manufacturers or mobile applications (see for example [10–13]). The objective of these devices is to help detect risky situations and encourage medical care when necessary. However, such devices have two limitations. Firstly, they are aimed at people already aware of sound risks, which limits their scope for prevention. Secondly, for diagnosis, they present a technical limit with the lack of a sound level calibration reference, in particular concerning online tests. Kimball [14] finds differences that can exceed 10 dB on average at certain frequencies between an online hearing assessment and a clinical assessment. Such online tests may therefore present a risk rather than a benefit, particularly for use in fitting hearing aids. Ultimately, this leads some authors to use additional equipment, such as digital-to-analog converters (DAC), to ensure the validity of the tests to the detriment of more generalized access to these devices (e.g. [10, 15]).

Taking these constraints into account beforehand makes it possible to consider the development of a hearing assessment device that is both reliable, in order to guarantee the validity of the tests, and affordable, in order to ensure its dissemination. Here, we present the Aupiometer, a low-cost and open-source audiometer, which was designed based on a Raspberry Pi nanocomputer. The hardware and functionalities of the device are first detailed. Then an evaluation of an automated test of standard and extended high-frequency PTA from 0.125 to 16 kHz is carried out in comparison with a clinical device, with the aim of certify the reliability of the Aupiometer. By taking advantage of its modularity and reliability over a wide frequency spectrum, the use of the Aupiometer can be extended to other hearing tests for a more complete hearing diagnosis.

## Operating principles of the Aupiometer

### Tools and developments

The Aupiometer was developed on a model 3+ Raspberry Pi with a 16 GB SD memory card (USD$70), its power supply (USD$10), a USB DAC DragonFly Red (USD$175), and Beyerdynamic DT-770 Pro 32 ohms headphones (USD$125). To guarantee the portability of the device, the Raspberry Pi was associated with a 7" touchscreen display Element14 (840x480; USD$80), both integrated in the same case (USD$15), a Rii i8 mini-keyboard and a touch pen (USD$20; cf. Fig 1). In total, the cost of the device is around USD$495. Note that in terms of manpower, for one test, it takes approximately one to two weeks full-time for computer development and calibration. For comparison, a clinical audiometer like the ELIOS from the company Echodia®, which serves as a reference in the experimental part, generally represents a cost greater than USD$1000 when limited only to the standard PTA. Moreover, the implementation of each new test and the associated equipment add a cost of the same order (e.g. USD$1000 for high-frequency audiometric headphones). Note that this cost difference can be partly explained by the regulatory "CE" marking for medical devices.

The Aupiometer modules were programmed using the free software Pure Data, installed natively on the Raspberry Pi with the Patchbox OS (48-kHz sampling frequency). The

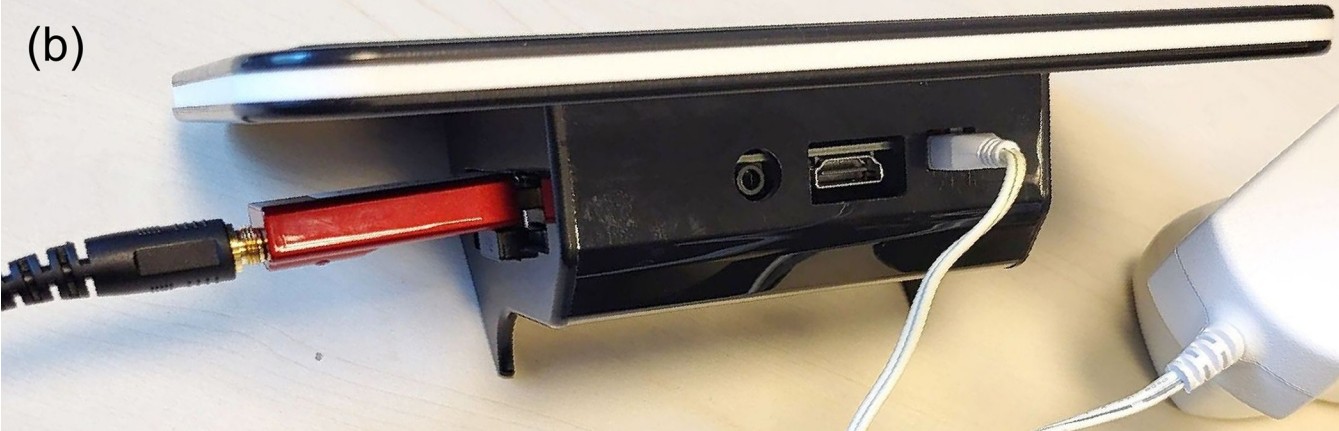

**Fig 1. Aupiometer equipment.** (a) Front view: the Aupiometer consists of a touch screen integrated into a Raspberry Pi, with which is associated a mini-keyboard, a touch pen, and headphones. (b) Rear view: the headphones are connected to a USB DAC on the back of the apparatus. The Aupiometer programs are freely available online.

programs developed as well as the installation procedure are available online in open-source (all codes are available at: https://doi.org/10.5281/zenodo.11033174, accessed on 22 April 2024). The programming of the modules and the complete configuration of the Aupiometer was carried out on a PC with an SD card reader before transferring them to the Raspberry Pi, although it is also possible to program directly on the Raspberry Pi. The Aupiometer installation procedure takes less than 15 minutes, from installing the OS to transferring the programs. After turning on the device, the program automatically starts on the Aupiometer home page to save the participant's information. For demonstration purposes, four modules have already been implemented: an automated standard and high-frequency PTA module, two speech audiometry modules and a hyperacusis questionnaire. Participant data and test results are saved as text files on the SD card, in the Aupiometer folder. The interfaces of the test modules are shown in Fig 2.

## Pure-tone audiometry module

The validity of the Aupiometer, in terms of calibration, reliability and use, is tested using the standard and high-frequency PTA. Indeed, this test covers a wide frequency spectrum and is presented with a dedicated interface, so as to constitute a good proof of concept for extending the functionalities of the Aupiometer to other hearing tests. More precisely, the PTA procedure used, which follows the Hughson & Westlake limits method, is fully automated for independent use by the participant (see also [19]). The procedure implemented on the Aupiometer is similar to that of the ELIOS clinical audiometer and complies with good practices [20]. Indeed, the test signal is a train of five beeps, each beep comprising a fade-in of 50 ms, a plateau of 200 ms, a fade-out of 50 ms, and a pause of 100 ms, i.e. a total duration of stimulation of 2000 ms. The starting sound level is set at 40 dB HL. If the participant presses the response button before the end of the stimulation, the sound level on the next trial is reduced by 10 dB, while if s/he does not press or if s/he presses too late, the sound level at the next trial is increased by 5 dB. The trials are separated by a pause between 2000 and 4000 ms randomly. However, the hearing threshold validation procedure differs between the two audiometers in two aspects. Firstly, when the participant presses the response button, the stimulation is stopped with the ELIOS device, while it continues until the end of 2000 ms with the Aupiometer in order to guarantee sufficiently long stimulation [20]. Secondly, for both devices, at the start of the test, when the participant correctly detects the stimulation and the sound level begins to decrease (i.e. from 40 dB HL to 30 dB HL), the threshold is then validated after two inversions of the sound level (i.e. an omission then a correct detection). However, when the sound level begins to increase at the start of the test (i.e. from 40 dB HL to 45 dB HL), the threshold is validated after a single inversion for the ELIOS (i.e. from the first correct detection), and three inversions for the Aupiometer in order to guarantee double validation.

The audiometric sound level intervals measurable with the Aupiometer depend on the signal amplification level (cf. Table 1). The choice was made to use the entire amplification interval of the digital signal from 0 to 100 dB in steps of 5 dB (where 0 dB corresponds to a RMS value of 0 in Pure Data, and 100 dB to a value of 1), in order to be able to generate the loudest possible signals before saturation in case of participants with severe hearing loss. In addition, at 0 dB the digital signal is zero, the minimum sound level is therefore emitted at 5 dB. At 100 dB, the digital signal emits the maximum sound level. If the participant does not press the response button at the maximum sound level for a given frequency, the Aupiometer records the corresponding value in dB HL plus 5 dB. The audiometric sound level ranges measurable with the Aupiometer are shown in Table 1, and correspond at least to the ranges of a type 4 audiometer according to standard EN606045-1.

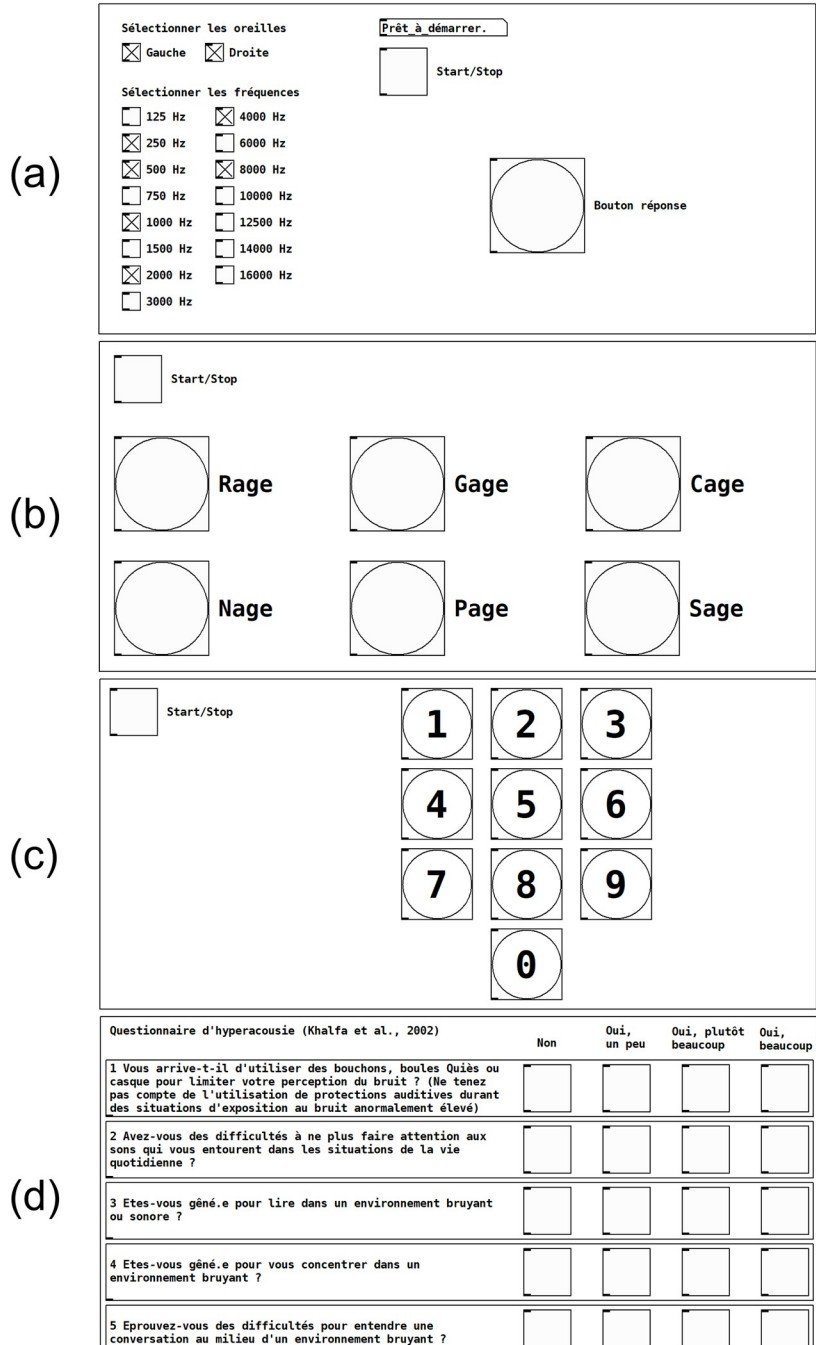

**Fig 2. Interfaces of the modules implemented on the Aupiometer.** (a) Standard and high-frequency PTA; (b) speech audiometry "Modified Rhyme Test" (MRT [16]); (c) speech audiometry "Digit Triplet Test" (DTT [17]); (d) hyperacusis questionnaire ([18]; note that the display is cut relative to the screen size, a side scroll bar allows to scroll through the rest of the questionnaire).

## Calibration module

An additional calibration module allows generating all the tested frequencies in order to check the corresponding sound level in dB SPL using an artificial ear. Table 1 indicates the final estimated Reference Equivalent Threshold Sound Pressure Levels (RETSPLs) of the Aupiometer

**Table 1. Calibration sound levels of the Aupiometer associated with Beyerdynamic DT-770 Pro 32 ohms headphones.** The calibration of an audiometer is usually carried out at 70 dB HL; the values listed in parentheses after the RETSPL values correspond to the measurements taken, averaged over the two headphones, with a digital amplification level fixed according to the last perceptual evaluation of this study. Audiometric level ranges range from 5 dB digital amplification to 100 dB plus 5 dB (see text). *The RETSPL levels of the frequencies 0.125, 14 and 16 kHz are measured at 100 dB before saturation without being able to reach 70 dB HL with the device used; the corresponding RETSPL values before the parentheses are estimated by adding the missing difference.

| Frequency (kHz) | RETSPL (dB SPL) | Signal amplification level (dB) | Audiometric sound level range (dB HL) |
|---|---|---|---|
| 0.125 | 37.2 (102.2*) | 35 (100 instead of 105*) | [–30 70] |
| 0.250 | 19.6 (89.6) | 30 (100) | [–25 75] |
| 0.500 | 12.5 (82.5) | 20 (90) | [–15 85] |
| 0.750 | 7.4 (77.4) | 15 (85) | [–10 90] |
| 1 | 7.5 (77.5) | 15 (85) | [–10 90] |
| 1.5 | 6.1 (76.1) | 15 (85) | [–10 90] |
| 2 | 7.9 (77.9) | 15 (85) | [–10 90] |
| 3 | 6.3 (76.3) | 15 (85) | [–10 90] |
| 4 | -3.9 (66.1) | 15 (85) | [–10 90] |
| 6 | 22.8 (92.8) | 20 (90) | [–15 85] |
| 8 | 23.0 (93.0) | 15 (85) | [–15 90] |
| 10 | 27.5 (97.5) | 25 (95) | [–20 80] |
| 12.5 | 26.7 (96.7) | 25 (95) | [–20 80] |
| 14 | 33.3 (98.3*) | 35 (100 instead of 105*) | [–30 70] |
| 16 | 52.8 (102.8*) | 50 (100 instead of 120*) | [–45 55] |

(i.e. obtained after the perceptual evaluation in comparison with the ELIOS and not via a standard normalization on the Aupiometer). Thus, the calibration offset in the PTA module can be readjusted if necessary depending on the equipment used. The increase of the sound level in steps of 5 dB was also verified using an artificial ear.

## Perceptual assessment of the pure-tone audiometry

### Participants

This study has received approval from the French Ethics Committee (CPP Tours–Région Centre–Ouest 1, n˚2020T2-15). Participants were recruited from July 27, 2023 to August 11, 2023. All participants provided written informed consent prior to any data collection. Four participants (including an author) were recruited for a pre-calibration test of the Aupiometer in order to make a first perceptual adjustment of sound levels by frequency (3 men and 1 woman; mean age 35.25 ± 2.99). Sixteen participants then carried out the perceptual evaluation following the same experimental procedure (5 men and 11 women; mean age 37.75 ± 8.23). All participants in the perceptual evaluation were normal hearing in the clinical sense, i.e. with thresholds below 25 dB HL from 0.250 to 8 kHz and for each ear, measured with the ELIOS.

### Apparatus

The perceptual tests are carried out with the Aupiometer and the Beyerdynamic DT-770 Pro 32 ohms headphones, on the one hand, and the ELIOS with the Radioear DD450 headphones, on the other hand. The evaluation covers standard and high-frequency PTA, knowing that the manufacturers' technical data sheets indicate a nominal frequency response between 0.005 and 35 kHz for the Beyerdynamic DT-770, while the Radioear DD450 is standardized for PTA from 0.125 to 16 kHz. Calibration of the Aupiometer was carried out using a Brüel & Kjær Type 2250 sound level meter and a Brüel & Kjær Type 4153 artificial ear connected to a G.R.A.S. Type 12AK power supply. Each audiometric device consists of a small touch screen with a

response button that can be pressed with a touch pen. Participants are placed in a double-walled IAC audiometric booth for evaluation with both devices.

The calibration of the PTA on the Aupiometer took place in three steps. The sound level of the Beyerdynamic DT-770 headphones connected to the Aupiometer was adjusted as a first approximation using the RETSPL values of the Radiohear DD450 headphones. Then, the data from four participants were collected following the procedure described in the following paragraph, identically to the perceptual evaluation. The average difference between the Aupiometer and the ELIOS, calculated on the average difference per frequency on the two ears, was -0.63 ± 3.76 dB. The deviation by frequency was rounded to the multiple of 5 in order to adjust the initial calibration for the perceptual evaluation. Finally, the calibration values of the Aupiometer presented in Table 1 are obtained after a final adjustment of ±5 dB on three frequencies after the perceptual evaluation (2, 10 and 16 kHz, for an average difference between frequencies of -0.26 ± 2.80 dB). The final difference between frequencies is estimated at -0.59 ± 1.14 dB.

## Procedure

The perceptual evaluation consisted of carrying out the PTA with the Aupiometer, on the one hand, and with the ELIOS, on the other hand, which served as a reference. The order of passage between the two tests was counterbalanced between participants. Frequencies tested ranged from 0.125 to 16 kHz, with 15 frequencies tested in the following order: 1, 1.5, 2, 3, 4, 6, 8, 10, 12.5, 14, 16, 0.750, 0.500, 0.250 and 0.125 kHz. As described previously, the stimuli and the test procedure were similar between the two devices (see paragraph Pure-tone audiometry module). The instructions given to participants were to respond by pressing the response button using the touch pen immediately after hearing the sound signal. In addition, before the test with each device, participants were told to be "fairly sure" of having heard the sound signal to respond. Participants took a short break between the two audiometry tests. Each audiometry test lasted approximately 20 min, for a total duration of the experiment for a participant of approximately 45 min.

## Statistical analyses

To study the effects of the different factors taken into account, we conducted several mixed analyzes of variance (ANOVA). Significant effects are analyzed with Tukey-HSD post-hoc tests. Furthermore, to analyze in more detail the agreement between the measurements taken with each of the two devices, a Bland-Altman analysis is carried out for each frequency on all 32 ears. The 90% limits of agreement are calculated (i.e. the interval within which any new measurement has a 90% chance of falling), which correspond to 1.64 standard deviations on either side of the bias. Furthermore, when mentioned, age is studied as a categorical variable by separating the participants into two groups depending on whether their age is strictly lower (N = 7) or greater than or equal to the median 37 y.o. (N = 9). Before running statistical tests, all assumptions they required were checked, concerning the normality of the data, the variance homogeneity between group and the sphericity of the data for within-subjects testing. A statistical test will be considered as significant when the p-value is less than 0.05. All statistical analyses were performed using Statistica software version 13 (TIBCO Software Inc. CA USA).

## Results

A first analysis concerned the order of passage between the two devices. A mixed ANOVA was carried out on the hearing thresholds with the within-subject variables of frequency, ear and device, and the between-subject variable of the order of passage. This analysis did not reveal

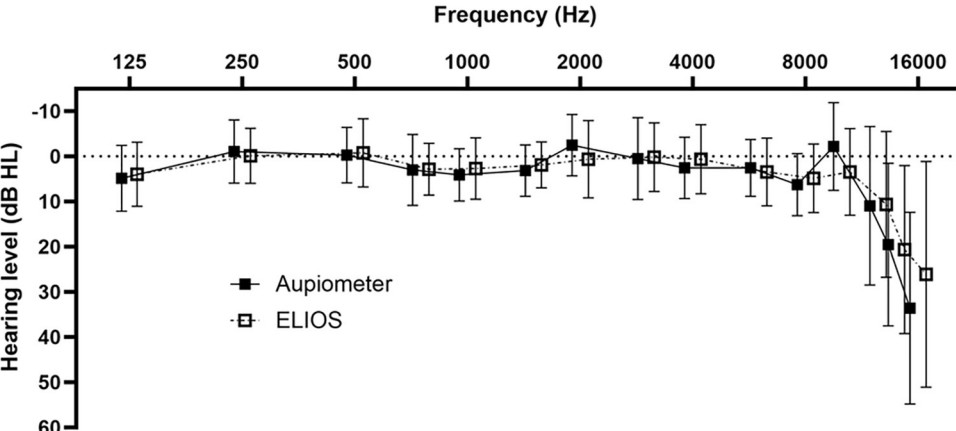

**Fig 3. Average audiograms obtained with the Aupiometer and the ELIOS on all 32 ears.** Error bars represent standard deviations.

significant main effect of the order of passage [$F(1,14) = 0.055$, $p = 0.819$, $\eta_p^2 = 0.004$], nor any significant interaction with frequency, ear or device [resp.: $F(14,196) = 0.361$, $p = 0.984$, $\eta_p^2 = 0.025$; $F(1,14) = 0.828$, $p = 0.378$, $\eta_p^2 = 0.056$; $F(1,14) = 3.048$, $p = 0.103$, $\eta_p^2 = 0.179$]. The order of passage therefore had no influence on the participants' performances and this factor was not taken into account in the following analyses.

Fig 3 represents the average of the audiograms performed on all 32 ears, for each device. The average hearing threshold measured with the ELIOS was 5.39 ± 13.36 dB HL, and 5.65 ± 13.95 dB HL with the Aupiometer. To determine if a difference exists on the measurement of hearing thresholds between the two devices, a mixed ANOVA was performed on the hearing thresholds with the within-subject variables of frequency, ear and device, as well as the between-subject variables of age and gender. The results did not indicate significant main effect of ear, device, age, or gender [resp.: $F(1,13) = 0.027$, $p = 0.872$, $\eta_p^2 = 0.002$; $F(1,13) = 0.055$, $p = 0.818$, $\eta_p^2 = 0.004$; $F(1,13) = 2.530$, $p = 0.136$, $\eta_p^2 = 0.163$; $F(1,13) = 3.390$, $p = 0.089$, $\eta_p^2 = 0.207$]. However, the analysis showed a significant main effect of frequency with an increase in thresholds at high frequencies starting at 12.5 kHz [$F(14,182) = 17.996$, $p < 0.0001$, $\eta_p^2 = 0.581$]. Indeed, the hearing thresholds did not present differences between all frequencies from 0.125 to 10 kHz [$p > 0.3$], while those at 14 and 16 kHz were significantly higher than all other frequencies [$p < 0.015$] including between them [$p < 0.007$]. Finally, at 12.5 kHz, the threshold was significantly higher than frequencies 0.25, 0.5, 2, 3, 4, 10 kHz [$p < 0.02$]. Furthermore, the interaction between frequency and age was significant [$F(14,182) = 9.198$, $p < 0.0001$, $\eta_p^2 = 0.414$]. The hearing thresholds of young participants showed no difference across all frequencies [$p > 0.2$], while the hearing thresholds of the elderly participants compared to the young ones were significantly higher at high frequencies at 14 and 16 kHz [$p < 0.0001$], which seems to indicate an effect of presbycusis. Moreover, the interaction between device and frequency was also significant [$F(14,182) = 7.099$, $p < 0.0001$, $\eta_p^2 = 0.353$]. Indeed, no significant difference was observed between the two devices as a function of frequency [$p > 0.4$], except at 10 and 16 kHz [$p < 0.0001$] (cf. Fig 3).

Across all hearing thresholds measured with the Aupiometer and the ELIOS, for each frequency, the differences between the two devices ranged between -40 and +20 dB (cf. Fig 4). In addition, 4.38% of these differences were greater than or equal to 15 dB in absolute value, which could therefore be considered as outliers values but which were kept in the rest of the analyses in order to prevent from additional bias. Analyses following the Bland-Altman

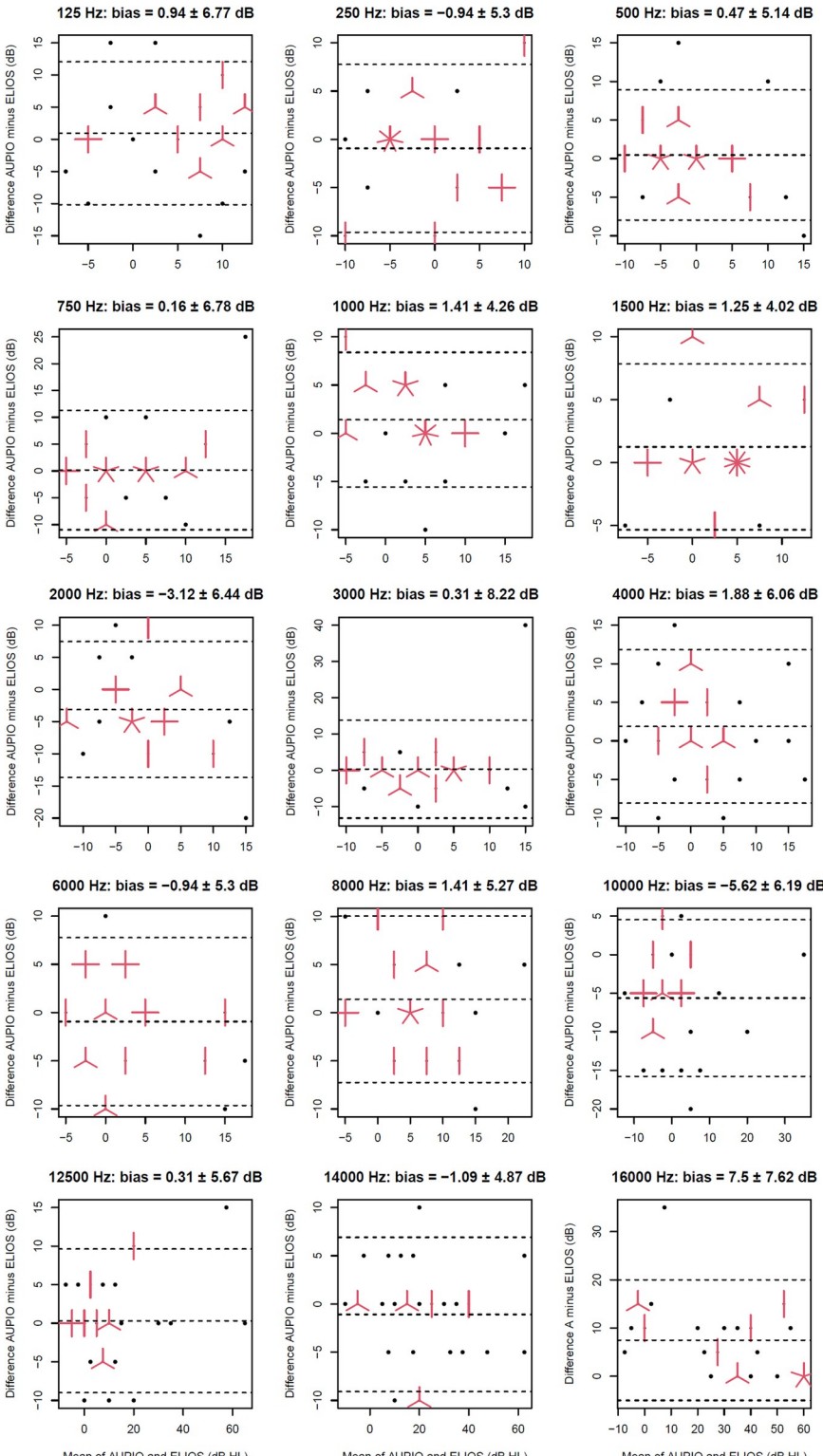

**Fig 4. Bland-Altman analyses for the comparison of the Aupiometer with the ELIOS.** Each panel corresponds to a given frequency and represents the bias and the 90% limits of agreement between the measurements taken with the two audiometers. Each point represents one ear and each red bar represents an overlapping value. The title of each panel indicates the value of the bias ± one standard deviation.

method indicated an average bias between frequencies of 0.26 ± 2.80 dB. The lower limit of agreement averaged -9.35 ± 3.11 dB, while the upper limit of agreement averaged 9.87 ± 3.67 dB. As mentioned previously, the bias observed at 2, 10 and 16 kHz was corrected afterwards in order to validate the final calibration of the Aupiometer (see paragraph Apparatus). Finally, concerning the two groups of young and elderly participants, the average bias between frequencies was 0.95 ± 3.82 dB for the young participants, with an average lower limit of agreement of -8.58 ± 4.86 dB and an upper limit of agreement of 10.48 ± 5.70 dB. For the elderly participants, the average bias between frequencies was -0.28 ± 2.26 dB, with an average lower limit of agreement of -9.36 ± 2.99 dB and an upper limit of agreement of 8.80 ± 3.39 dB. The effect of age-related hearing loss therefore did not seem to induce an increase in the variability of responses between the two devices, including at high frequencies, as observed in Fig 4.

## Discussion

### Validation of the Aupiometer

In this study, we introduced the Aupiometer, a low-cost, open-source and modular audiometer. Its validity was checked by comparison with a clinical audiometer, to consider clinical use or as an education or research tool. It has been done with normal-hearing participants (up to 8 kHz), in order to verify the valid correspondence with a clinical audiometer around the reference value of 0 dB HL. However, some participants presented high-frequency losses with thresholds of up to 65 dB HL, which were verified on both devices. Thus, the calibration carried out to reach the reference values across the frequencies, as well as the validity of the digital amplification of the signal in the measurement intervals, makes it possible to extend the use of the Aupiometer to participants with hearing losses. Also note that the Aupiometer has been validated here to reproduce a clinical situation, with closed headphones and in an audiometric booth to guarantee the sound insulation of the participants and not introduce bias into the verification of the calibration. Depending on the context of use, it may be recommended to use headphones with better sound insulation than the ones associated here with the Aupiometer, in particular if the tests are done outside an audiometric booth.

After the final calibration over the entire wide spectrum of frequencies tested from 0.125 to 16 kHz, the average difference between the two devices and between frequencies is estimated at only -0.59 ± 1.14 dB, for an average standard deviation between frequencies estimated at 5.86 ± 1.18 dB. These differences are in the range of test-retest differences on a single device measured in previous studies. Indeed, the order of magnitude of the tolerated variability which seems to be a consensus is ±10 dB for a 90% confidence interval (e.g. [21]). With the Aupiometer, the variability is ±9.61 dB following this criterion. In addition, most of the studies in the literature concern standard PTA, i.e. from 0.250 to 8 kHz. Stuart et al. [22] present for example mean differences of 1.58 ± 4.53 dB for young adult participants in manual PTA. Here, for the same frequencies, mean differences between the two devices are -1.02 ± 5.41 dB. Likewise, in automated standard PTA between 0.500 and 6 kHz, the meta-analysis by Mahomed et al. [23] indicates mean differences of 0.3 ± 6.9 dB, while our results indicate mean differences of -0.83 ± 5.90 dB. In all cases, our results are of the same order of magnitude as those observed in the literature on test-retest PTA.

Moreover, according to our analyses, it does not appear that moderate hearing losses for elderly participants induce an increase in the observed variability. These results are consistent with those of Hoff et al. [19], who studied the variability of responses to standard PTA in two groups of elderly participants (70 and 85 years old). Comparing an automated test to a manual test, the authors observe good data replication with approximately 90% of the data falling within the ±10 dB interval, and this variability does not appear to be predominantly associated

with age, gender, auditory state or cognitive state. Swanepoel et al. [9] also obtained good replication of data in automated standard PTA, comparable to its manual version, for both normal-hearing and hearing impaired participants. Thus, it appears that automated PTA, such as that implemented in the Aupiometer and including in high frequencies, presents good reliability whatever the hearing state.

Yet, other factors can have increased the differences between the two successive tests with the Aupiometer and the clinical device, such as motivation, attention or fatigue of the participant, due to the higher number of frequencies tested than usual. To prevent from additional bias, all data were analyzed without excluding outlier data, although certain extreme values could greatly increase the variance of the data collected. Technical factors linked to the Aupiometer can also explain part of this variability. In particular, with the choice of blocking the increment of the sound level between 0 and 100 dB with multiple amplification steps of 5 dB, rather than calibrating the sound level to the nearest dB. This choice was made in order to use all the digital signal amplitude available, and thus be able to generate the maximum sound level with the equipment used if necessary, although the maximum thresholds that can be measured at low frequencies (70 dB HL at 125 Hz) as well as at high frequencies (55 dB HL at 16 kHz) remain relatively low. However, the maximum thresholds are sufficient to detect severe hearing loss at low as well as high frequencies (e.g. 80 dB HL at 12.5 kHz), and using an external amplifier would remove this limitation without greatly increasing the cost of the device. Finally, the sound levels correspond at least to a type 4 audiometer according to standard EN606045-1 (which only takes into account frequencies between 0.250 and 6 kHz).

## Applications

Several recent studies proposed online or mobile application audiometric tools. The online standard PTA presented by Renganath & Ramkumar [15] nevertheless requires a specific sound card and connector, in addition to a computer connected to the internet to allow remote data transmission. The application developed by Sørensen et al. [10] allows automated standard PTA to be performed on a smartphone but also requires an external DAC like the one used in our study. Furthermore, prototypes on Raspberry Pi (e.g. [24]) or Arduino (e.g. [25]) have been proposed, but without audiological validation. Thus, it seems difficult to guarantee the good reproducibility of the tests without certain material constraints. Nonetheless, the Aupiometer aims to overcome these limits by offering a single integrated and modular device with audiological validation, essential whatever the considered use cases (e.g. education, clinical, academic research).

The Aupiometer may be of interest to the world of education in order to extend access to training tools (e.g. audiologist training), or by integrating into self-service devices to reach populations of different age groups (e.g. universities, workplaces), or in teleconsultation particularly when far from medical centers (e.g. rural areas, developing countries). For demonstration purposes and as a first approximation, the system can be used on the same hardware references with the same calibration values; for precision uses, calibration check will be necessary using an artificial ear, by referring to the RETSPL values specified here. Its modules can be adapted to each of these use cases. In particular, extended high-frequency PTA as proposed on the Aupiometer is rarely offered on clinical devices, even though it provides a good indicator of hearing state [26]. However, PTA is not sufficient to precisely characterize the hearing state or it can even be misleading in case of synaptopathy [27]. It should therefore be supplemented by other tests which are more relevant to quantify the hearing complaint, such as speech audiometry in noise. Today, on the Aupiometer, it is already possible to implement various speech audiometry tests (e.g. MRT, DTT) as well as questionnaires (e.g. hyperacusis

questionnaire). The integration of other hearing tests is considered, such as other methods of measuring PTA (e.g. Audioscan, Békésy [28]), and also by bone conduction with a vibrator to separate sensorineural vs. conductive hearing loss; other speech audiometry tests (e.g. Coordinate Response Measure [29]); a tinnitus frequency recognition test or questionnaire (e.g. Tinnitus Handicap Inventory [30]); noise exposure questionnaires (e.g. [31]); or even tests more complex as otoacoustic emissions to assess objectively the cochlear amplification function with an otoacoustic-emission probe [32].

Finally, making this device available to research teams would also be a good way to generalize and standardize the monitoring of the hearing of participants in hearing and cognitive studies, and to develop new tests, while certain research teams sometimes use custom audiometry assessment (e.g. [33]). Additionally, the use of automated tests as implemented in the Aupiometer could reduce experimental bias. For example, when including a large number of participants evaluated by several experimenters, or for longitudinal monitoring by providing each participant with a copy of this affordable device. The collection of data to investigate the causal relationships between exposure to noise and hearing and cognitive impairments could thus be improved.

## Conclusion

The results obtained using the Aupiometer in standard and extended high-frequency PTA made it possible to validate this low-cost and open-source portable audiometer as a reliable audiometer that complies with audiological requirements. The device is easily mastered by the participants for use in automated mode, which also facilitates data collection by limiting the bias linked to the experimenter. Thus, the Aupiometer is likely to generalize access to hearing assessment. Finally, taking advantage of its modularity, this device is also intended for academic research, with the possibility of programming and configuring innovative tests adapted to experimental problems, and of more easily deploying several devices to follow a larger number of participants over time for longitudinal studies.

## Acknowledgments

We thank Jean-Christophe Bouy, Grégory Gérenton and Romain Moura for their scientific and technical advice.

## Author Contributions

**Conceptualization:** Vincent Isnard, Guillaume Andéol.

**Data curation:** Vincent Isnard.

**Formal analysis:** Vincent Isnard, Véronique Chastres, Guillaume Andéol.

**Investigation:** Vincent Isnard.

**Methodology:** Vincent Isnard.

**Project administration:** Vincent Isnard, Véronique Chastres, Guillaume Andéol.

**Resources:** Vincent Isnard.

**Software:** Vincent Isnard, Véronique Chastres.

**Supervision:** Guillaume Andéol.

**Validation:** Vincent Isnard, Véronique Chastres, Guillaume Andéol.

**Visualization:** Vincent Isnard, Véronique Chastres.

**Writing – original draft:** Vincent Isnard.

**Writing – review & editing:** Vincent Isnard, Véronique Chastres, Guillaume Andéol.

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
