## [Decision Letter · Decision Letter 0]

12 Mar 2024

PONE-D-24-01234Description and validation of a new low-cost and open-source audiometer: The AupiometerPLOS ONE

Dear Dr. Isnard,

Thank you for submitting your manuscript to PLOS ONE. After careful consideration, we feel that it has merit but does not fully meet PLOS ONE’s publication criteria as it currently stands. Therefore, we invite you to submit a revised version of the manuscript that addresses the points raised during the review process.

We look forward to receiving your revised manuscript.

Kind regards,

Paul H Delano, Ph.D.

Academic Editor

PLOS ONE

2. We note that Figure 1 in your submission contain copyrighted images. All PLOS content is published under the Creative Commons Attribution License (CC BY 4.0), which means that the manuscript, images, and Supporting Information files will be freely available online, and any third party is permitted to access, download, copy, distribute, and use these materials in any way, even commercially, with proper attribution. For more information, see our copyright guidelines: http://journals.plos.org/plosone/s/licenses-and-copyright.

Additional Editor Comments:

Highlight in the title that this technology has been tested only in normal hearing individuals.

Would it be possible to separate sensori-neural versus conductive hearing loss?

Reviewers' comments:

Reviewer's Responses to Questions

**Comments to the Author**

1. Is the manuscript technically sound, and do the data support the conclusions?

Reviewer #1: Yes

Reviewer #2: Yes

2. Has the statistical analysis been performed appropriately and rigorously? 

Reviewer #1: Yes

Reviewer #2: Yes

3. Have the authors made all data underlying the findings in their manuscript fully available?

Reviewer #1: Yes

Reviewer #2: No

4. Is the manuscript presented in an intelligible fashion and written in standard English?

Reviewer #1: Yes

Reviewer #2: Yes

5. Review Comments to the Author

Reviewer #1: The work presents a new open-source audiometer based on a Raspberry Pi single-board computer. The manuscript describes the design details and validates it.

The paper is well-written and clear. The methodology is sound, and the results are interesting. In my opinion, the disclosure of this design is a contribution to the scientific community. It is important to highlight that the software is open-source and is available on GitHub. For these reasons, I recommend accepting the paper after the following issues are addressed:

* Major Issues

- There are missing elements in the determination of the device's cost in order to make a fair assessment and comparison of its cost. In particular, you need to include an estimation of labor cost and the calibration cost. I also miss the cost of the Raspberry case and the power supply. Along the same lines, please include in the photo of Figure 1 a view where one can observe the whole device, including the Raspberry Pi with its case and power supply.

- Regarding the experiments, clarify if all sixteen participants have normal hearing or not. If yes, is this a limitation of the study? Should future work include testing the device with participants with hearing loss?

- The experiments were performed in an audiometric booth. I understand this is a methodological requirement in the context of this manuscript. However, it is not clear to me if an audiometric booth is always required. Please include a discussion about this issue. In particular, clarify if using the device without the booth is possible. Moreover, if possible, state under which circumstances it is possible and under which limitations.

* Minor Issues

- Line 32: Replace "cognitive decline" with "and cognitive decline."

- Line 54: The writing of the sentence "automated and calibrated, designed based on a Raspberry Pi nanocomputer." is awkward. Please fix.

- Line 58: The writing of the sentence "it is a question of considering the use of the Aupiometer for a broader diagnosis including other hearing tests." is awkward. Please fix.

- In the paragraph starting at line 63, use "USD$" instead of "$" (assuming prices are in US dollars).

- Line 77: Explain what is "Fe" (I assume it is the sampling frequency, but please be explicit).

- Link your GitHub to a Zenodo repository (https://zenodo.org/). The service is free, you will get a DOI, and you can add your repository to your references.

Reviewer #2: The article “Description and validation of a new low-cost and open-source audiometer: The Aupiometer” propose a new device integration (raspberryPI + headphone + software) to performa PTA, and a validation comparing the new device with a clinical standard one.

In general, the article is clearly written, but it requires further explanations as it is intended as an OpenSource PTA.

Major comments

Main motivation is to propose a low-cost device, then authors indicate that the final cost is about 50% of a commercial setup (main cost is headphone). For me 400 US$ is not low cost and 50% is not a huge price reduction, but at the same time PTA is usually performed is silent rooms, by specialized personnel, so comparison is not fair. Also, it is necessary the expensive headphone or the newer raspberryPI? Please clarify the argumentation.

Related with evaluation, were both devices evaluated in an isolated room? How sensitive to noise conditions and personnel are the measurements? How was the perception of the device by the uses? (usability of the system).

The proposed device requires a calibration step, which is standard, but that makes hard to think in telemedicine applications as that requires special equipment. Have the authors thought on how to address this issue?

I checked the git repository and did not find the data tables, just code. Could the authors provide the data of comparison?

Minor comments

Abstract should include main results (dB mean difference, N subjects, frequency range).

Figures are low resolution. Figure 1, screen brightness prevents getting an idea of the software. Figure 2, is that really the interface or the mockup? Please provide final interface or at least a mockup with good resolution. Figure 4, I cannot read the legends (make plot vectorial for clear reading)

6. PLOS authors have the option to publish the peer review history of their article (what does this mean?). If published, this will include your full peer review and any attached files.

Reviewer #1: **Yes: **Alejandro Weinstein

Reviewer #2: No

---

## [Author Response · Author response to Decision Letter 0]

9 May 2024

Dear Editor, dear Reviewers,

We thank you for all the comments which greatly helped to improve the quality of the manuscript. We have made corrections for each of these comments, which we detail below.

- We have corrected the format of the manuscript to match the style requirements.

- With the response provided (“we can proceed without any edits to figure 1”) following the request for clarification that we sent on 4/16/2024, Figure 1 is maintained. Modifications are nevertheless made to this figure following comments from Reviewers.

Additional Editor Comments:

- The title was modified accordingly.

- Yes, it would be possible to separate sensori-neural versus conductive hearing loss by performing a pure-tone audiometry measurement by bone conduction. We added mention of this possibility under discussion (l. 378, in the revised manuscript with track changes).

Reviewer #1:

• Major issues:

- Costs:

o Indeed, it was important to add an estimate of computer development time (l. 87). However, we have not specified an associated monetary cost due to salary disparities between countries.

o You are right, the cost of the Raspberry case and the power supply were missing and have been added (l. 83).

o A photo of the rear view of the device has been added to Fig 1.

- Participants:

o Yes, all sixteen participants have normal hearing, this is now clarified (l. 189).

o We discussed the validity of the calibration for participants with hearing loss (l. 302).

- We discussed the use of an audiometric booth (l. 273).

• Minor issues:

- L. 32: correction made.

- L. 54: correction made.

- L. 58: correction made.

- Paragraph l. 63: correction made.

- L. 77: correction made.

- Zenodo repository: a link has been added (l. 102).

Reviewer #2:

• Major comments:

- L. 82: indeed, the orders of magnitude of the costs referred to different tests and did not allow a comparison between the devices. We have added an example of hardware cost to give a better point of comparison (e.g. USD$1000 for high-frequency audiometric headphones; while the cost of implementing tests depends on the manufacturers).

- The evaluation was carried out in an audiometric booth, as was mentioned in the manuscript, we have clarified this better (l. 202). In addition, to complete the evaluation, participants only had to press a button on the touch screen of both devices (see “Procedure” paragraph), i.e. with very intuitive usability. However, usability with other naive experimenters was not evaluated, but it was designed to be equivalent to standard audiometric devices with a simplified display.

- We propose uses of the device in telemedicine for example in workplaces or other public spaces where the calibration of the device could be carried out, because it is indeed necessary to be equipped. We have clarified this (l. 366).

- The raw data table has been added to the computer code (with a new DOI link; l. 102), thank you for pointing this out to us.

• Minor comments:

- The abstract has been completed with the main results (l. 32).

- The quality of Figure 1 has been improved.

- Figure 2 is the actual interface as it appears on the device. The font used on the device is simple. It was specified in the legend that the display of the questionnaire is cut off by the size of the screen, but that the user has a scroll bar (l. 100).

- The quality of Figure 4 has been improved to make the legends readable.

Sincerely,

Vincent Isnard

---

## [Decision Letter · Decision Letter 1]

18 Jun 2024

PONE-D-24-01234R1Description of a new low-cost and open-source audiometer and its validation with normal-hearing listeners: The AupiometerPLOS ONE

Dear Dr. Isnard,

Thank you for submitting your manuscript to PLOS ONE. After careful consideration, we feel that it has merit but does not fully meet PLOS ONE’s publication criteria as it currently stands. Therefore, we invite you to submit a revised version of the manuscript that addresses the points raised during the review process.

(1) Include the cost of the Raspberry itself (only the cost of the display is included)

(2) Send Figures 1, 2, 4 with high resolution or vectorial file

We look forward to receiving your revised manuscript.

Kind regards,

Paul H Delano, Ph.D.

Academic Editor

PLOS ONE

Journal Requirements:

**Additional Editor Comments:**

All major concerns have been addressed.

A couple of minor changes:

(1) Include the cost of the Raspberry itself (only the cost of the display is included)

(2) Send Figure 1, 2, 4 with high resolution or vectorial

Reviewers' comments:

Reviewer's Responses to Questions

**Comments to the Author**

1. If the authors have adequately addressed your comments raised in a previous round of review and you feel that this manuscript is now acceptable for publication, you may indicate that here to bypass the “Comments to the Author” section, enter your conflict of interest statement in the “Confidential to Editor” section, and submit your "Accept" recommendation.

Reviewer #1: All comments have been addressed

Reviewer #2: All comments have been addressed

2. Is the manuscript technically sound, and do the data support the conclusions?

Reviewer #1: Yes

Reviewer #2: Yes

3. Has the statistical analysis been performed appropriately and rigorously? 

Reviewer #1: Yes

Reviewer #2: Yes

4. Have the authors made all data underlying the findings in their manuscript fully available?

Reviewer #1: Yes

Reviewer #2: Yes

5. Is the manuscript presented in an intelligible fashion and written in standard English?

Reviewer #1: Yes

Reviewer #2: Yes

6. Review Comments to the Author

Reviewer #1: All the comments have been addressed. I just noticed that the cost of the Raspberry itself is not included (only the cost of the display is included). Please add it.

Reviewer #2: The revised version of the article “Description of a new low-cost and open-source audiometer and its validation with normal-hearing listeners: The Aupiometer” made mainly clarifications about costs and methodology to compare with gold standard.

All major comments have been addressed.

About minor comments I still believe images are 1, 2, 4 are of poor resolution, and should be included vectorial or in high resolution.

7. PLOS authors have the option to publish the peer review history of their article (what does this mean?). If published, this will include your full peer review and any attached files.

Reviewer #1: **Yes: **Alejandro Weinstein

Reviewer #2: No

---

## [Author Response · Author response to Decision Letter 1]

19 Jun 2024

Dear Editor, dear Reviewers,

All points of correction have been addressed and commented in the response document.

Sincerely,

Vincent Isnard

---

## [Editor Report · Decision Letter 2]

23 Jun 2024

Description of a new low-cost and open-source audiometer and its validation with normal-hearing listeners: The Aupiometer

PONE-D-24-01234R2

Dear Dr. Isnard,

We’re pleased to inform you that your manuscript has been judged scientifically suitable for publication and will be formally accepted for publication once it meets all outstanding technical requirements.

Kind regards,

Paul H Delano, Ph.D.

Academic Editor

PLOS ONE
---

## [Editor Report · Acceptance letter]

1 Jul 2024

PONE-D-24-01234R2 

PLOS ONE

Dear Dr. Isnard, 

I'm pleased to inform you that your manuscript has been deemed suitable for publication in PLOS ONE. Congratulations! Your manuscript is now being handed over to our production team.

Kind regards, 

on behalf of

Dr. Paul H Delano 

Academic Editor

PLOS ONE